# Enhanced Adsorption of Methylene Blue Using Phosphoric Acid-Activated Hydrothermal Carbon Microspheres Synthesized from a Variety of Palm-Based Biowastes

Saeed Alhawtali [1], Mohanad El-Harbawi [1,*], Abdulrhman S. Al-Awadi [2], Lahssen El Blidi [1], Maher M. Alrashed [1] and Chun-Yang Yin [3]

[1]  Department of Chemical Engineering, King Saud University, Riyadh 11421, Saudi Arabia; saeerr2014@gmail.com (S.A.); lelblidi@ksu.edu.sa (L.E.B.); mabdulaziz@ksu.edu.sa (M.M.A.)
[2]  K.A. CARE Energy Research and Innovation Center in Riyadh, King Saud University, Riyadh 11421, Saudi Arabia; alawadi@ksu.edu.sa
[3]  Newcastle University in Singapore, 537 Clementi Road #06-01, SIT Building @ Ngee Ann Polytechnic, Singapore 599493, Singapore; chunyang.yin@newcastle.ac.uk
*   Correspondence: melharbawi@ksu.edu.sa

**Abstract:** In the present study, the ability for novel carbon microspheres (CMs) derived from date palm (*Phoenix dactylifera*) biomass using a hydrothermal carbonization (HTC) process and activated using phosphoric acid to remove methylene blue dye was investigated. Three types of palm-based wastes (seeds, leaflet, and inedible crystallized date palm molasses) were used and converted to CMs via the HTC process. The prepared samples were then activated using phosphoric acid via the incipient wetness impregnation method. The CMs samples before and after activation were analyzed using scanning electron microscopy (SEM), elemental analysis and scanning (CHNS), and the Fourier transform infrared (FTIR) and Brunauer–Emmet–Teller (BET) methods. The samples exhibited high BET surface areas after activation (1584 m$^2$/g). The methylene blue adsorption results showed good fitting to the Langmuir, Fruendlich, and Temkin isotherm models for all activated samples. The maximum adsorption capacity achieved was 409.84 mg/g for activated CM obtained from the palm date molasses, indicating its high potential for application as a dye-based adsorption material.

**Keywords:** date palm biomass; carbon microspheres; phosphoric acid activation; methylene blue; adsorption

## 1. Introduction

Hydrothermal carbonization (HTC) is an environmentally friendly, cost-effective, and efficient process for converting agricultural waste into valuable porous carbon (hydrochar). In recent years, hydrochars have received growing attention due to their low production costs and potential applications in many industrial fields. The synthesized hydrochars are generally manifested in the form of carbon microspheres (CMs). Carbon microspheres/hydrochars can be synthesized via a two-step HTC process: (1) the dehydration of carbohydrates to form a furan-like molecule (either 5-hydroxymethyl-2-furaldehyde or furfural) and (2) the subsequent condensation and polymerization of the furan compound(s) to form carbon materials [1,2]. In general, CMs can be produced by digesting various organic feedstocks, such as biomass or saccharides, at moderate temperatures (160–250 °C) and pressures. This approach is simple, straightforward, environmentally friendly, and cost-effective. Many researchers have prepared CMs/hydrochar using different materials, such as agricultural biomass [3,4], sewage sludge [5,6], food waste [7,8], saccharides [9,10], algae [11], and date palm molasses [12]. Hydrochar has been proven to be very effective in various industrial applications, especially in the adsorption of toxic substances from water, such as dyes, pigments, and heavy metals [3,13]. Various wastewater treatment processes,

including adsorption, membrane filtration, chemical precipitation, coagulation, ion exchange, electrochemistry, and electrochemical removal, can remove dyes and heavy metals from wastewater [14,15]. Some of these methods have substantial downsides, including high capital expenditure, high energy requirements, high maintenance and operation costs, partial removal, time-consuming regeneration, and the formation of toxic sludge [16,17]. Adsorption is one of the most effective and economical techniques for removing water contaminants [18].

In line with the global awareness of sustainable development, many researchers have worked to sustainably convert waste from various sources into useful materials. Therefore, it is favorable to repurpose these agricultural wastes, convert them into carbon-concentrated materials, and apply them in wastewater treatment. The use of date palm-based waste in wastewater treatment is, therefore, a two-pronged solution—it solves the issue of the abundance of palm wastes without burning them indiscriminately, and it converts them into useful adsorbent materials for the removal of toxic pollutants, such as dye contaminants, from wastewater.

In the Kingdom of Saudi Arabia, there are more than 31 million date palms [19], and each palm produces approximately 20 kg of waste per year [20]. The total waste of date palms is about 620,000 MT. Several attempts have been made to convert palm date biomass into valuable materials, such as activated carbon [21], hydrogel [22], biodiesel [23], and syngas [24]. However, most date palm wastes, such as leaves, twigs, and fibers, are not fully utilized in most cases and are usually burned, leading to numerous environmental problems (e.g., the unrestricted release of hazardous flue gas consisting of soot, $CO_2$, $NO_x$, $SO_x$, etc.).

The annual production of synthetic dyes is increasing worldwide as the demand for consumer goods increases. It has been estimated that textile mills currently produce more than 10,000 different types of dyes and pigments, with an approximated annual production of about $7 \times 10^5$ tons [25]. About 10–15% of this amount is discharged directly into the environment in the form of wastewater [26,27]. As the consumption of dyes is rapidly increasing due to high demand in various sectors, especially in the textile industry, the production of dyes is expected to increase, which will certainly lead to an increase in waste generated from the textile industry. Dye concentrations in textile effluents have been reported in a wide range of values (10–800 mg/L) [28].

Hydrochar can be synthesized at mild reaction conditions [29], which makes its production cost-effective. In addition, the presence of oxygen functional groups, such as hydroxyl, phenol, carbonyl, and carboxyl, on the surface of hydrochar enables the material to have an enhanced pollutant sorption ability [30]. On the other hand, the main disadvantages of hydrochar are its poor surface area and porosity, which hinder its ability to adsorb contaminants. Therefore, increasing their surface area and porosity can increase the adsorptive effectiveness of CMs. In previous studies, researchers concluded that activated hydrochar prepared from biomass waste had a much better adsorption capacity than non-activated hydrochar. Islam and co-researchers [31] synthesized activated hydrochar from coconut shells and sodium hydroxide and obtained a methylene blue (MB) adsorption capacity of 200.01 mg/g. Tran and co-researchers [32] prepared activated hydrochar using potassium hydroxide from coffee husk waste and obtained a maximum adsorption capacity of 418.8 mg/g for MB. Zhou and co-researchers [33] activated hydrochar made from sugarcane bagasse using phosphoric acid and sodium hydroxide. A maximum adsorption capacity of MB, 357.14 mg/g was obtained.

As far as we know, there is no known investigation focused on the use of a variety of date palm-based wastes for the synthesis of phosphoric acid-activated CMs in applications for dye-based wastewater treatment. It should be noted that these different types of date palm-based wastes are not homogenous, and their hydrothermal carbonization processes need to be properly optimized—this represents a novel aspect of the field of HTC. The findings from such a study would bode well for the sustainable use and repurpose of date palm-based wastes, which reflects our underlying motivation. We prepared three

samples of CMs from several types of biomass precursors of date palm (*Phoenix dactylifera*), namely, seeds, leaflets, and inedible crystallized date palm molasses. The prepared samples were subsequently activated with phosphoric acid to develop enhanced adsorption characteristics for enhanced methylene blue removal.

## 2. Materials and Methods

### 2.1. Raw Materials and Chemicals

Date palm seeds and leaflets were obtained from a local farm located on the outskirts of Riyadh, Saudi Arabia, while inedible crystallized palm molasses were obtained from a date factory located in Riyadh. The seeds and leaflets were washed several times with tap water to remove dust and dirt. The material was then dried in a drying oven at 80 °C for 48 h. The material was then ground into a powder using a Fritsch Pulverisette 15 cutting mill, Germany, equipped with a 0.25-mm sieve. Finally, the powdered material was sent to undergo the HTC process described in our previous studies [3,12]. Deionized (DI) water was utilized to clean the solid samples and for the HTC and dye adsorption processes. MB (purity 95%) was obtained from LOBA Chemie, India. Absolute ethanol was obtained from VWR, Spain, and used only for washing the solid materials prepared via the HTC process.

### 2.2. Hydrothermal Carbonization

An amount of 4 g of seeds, 2.5 g of leaflets and 4 g of inedible crystallized date palm molasses were placed in Erlenmeyer flasks with deionized water (25 mL) and magnetically stirred for 5 h. The samples were then placed in 45 mL Teflon-lined Paar reactors for HTC. The Paar vessels were hermetically sealed and heated in a muffle oven at 230 °C for 4 h with seeds, 3 h with inedible crystallized molasses, and 8 h with leaflets. The reason for the different reaction times is attributed to the chemical compositions of the parts, and the different times reflect the difficulty in biomass digestion processes associated with different types of wastes. It was found that molasses that contain about 47% glaucous [34] and seeds were relatively easier to convert to CMs than leaflets because they require lower temperatures and reaction times [35,36]. This could be due to the high cellulose content in leaflets (47.14% cellulose, 36.73% lignin, and 16.13% hemicellulose) [37]. In contrast, the seeds and date molasses contain (32.77% cellulose, 30.20% hemicellulose, and 37.03% lignin) [37] and (47.1% glucose, 28% fructose, and 4.7% sucrose), respectively [34]. The dark-colored liquids were filtered to produce solid products of HTC. They were then washed a few times with DI water and absolute ethanol. The wet products were dried at a temperature of 110 °C for 12 h in an oven. Subsequently, the dry final products were ground with a laboratory mortar and then stored in sealed bottles in a desiccator prior to the activation process.

### 2.3. Activation after HTC

The samples were activated with phosphoric acid according to the method of incipient wetness impregnation [38,39]. The effects of the phosphoric acid/sample ratio and activation temperature were studied. One gram of the CM sample prepared during the HTC process was impregnated via the dropwise addition of phosphoric acid solution. The sample was vigorously stirred until a thin liquid film of phosphoric acid formed on the sample surface. The effects of phosphoric acid concentrations and activation temperatures were investigated. The following ratios of phosphoric acid/CM sample were investigated: 0.5, 0.75, 1, 1.3, 1.7, 2.3, 3, 4, and 5. The impregnated sample was then dried overnight at 80 °C. Then, the impregnated sample was carbonized in a horizontal tube furnace (OTF-1200X-S, MTI, CA, USA) at three different temperatures (450, 500, and 650 °C) and an $N_2$ flow of 50 CCM for 1.5 h. The obtained products were washed several times with deionized water to remove any residual phosphoric acid, filtered off, and dried at 80 °C for 24 h. The samples are referred to as "NAS" (non-activated seed), "AS" (activated seed), and "AAS" (activated seed after adsorption). Finally, the prepared samples were stored in a dry place before being used for dye removal.

### 2.4. Characterizations

The absorbance rates before and after activation were characterized via scanning electron microscopy (SEM) using the Tuscan VEGA II LSU (Tuscan Inc., Addison, IL, USA) to study the surface morphology of the prepared materials. The ImageJ software (version 1.53t) was used to determine the diameter of CMs. The adsorbents were further characterized using the Micromeritics TriStar II PLUS (Norcross, GA, USA) surface analyzer to evaluate their surface area, pore diameter, and pore volume. The samples were degassed for 90 min at 250 °C under a vacuum. The data obtained from this analysis, which included pore size and surface area, were further analyzed and compared to determine changes in the surface characteristics of the CMs due to various chemical treatments and their effects on dye removal. A Fourier transform infrared (FTIR) analysis was performed on the three samples before and after activation. The FT-IR spectra are important as they indicate the presence of bonds on the microspheres, such as C–H, C=O, etc. The CMS samples were mixed with KBr to form a pallet for the FT-IR analysis. Wavelength ranges in the infrared region adsorbed by the adsorbents were measured using an FTIR spectrophotometer (model: Shimadzu IRPrestige-21, Tokyo, Japan). An elemental analysis of the samples was conducted via a PerkinElmer series II CHNS/O 2400 analyzer (VELP, Wisconsin, NY, USA). Oxygen content was established via the calculation of the remaining mass. Boehm titration [40,41] was performed on the produced samples to quantify the acidic (oxygenated) functional groups on the surface of inactivated and activated samples.

### 2.5. Adsorption Studies

The adsorptive abilities of the inactivated and activated prepared CMs were tested for MB removal using a batch adsorption series. The effects of the following parameters on the adsorption process were evaluated: pH, contact time, adsorbent dosage, and initial dye concentration. A stock solution (1000 mg/L of MB) (95% purity) was produced by dissolving MB in deionized water. Sodium hydroxide or sulfuric acid (0.1 mol/L) was used to adjust the pH and maintain it at 6. The pH of the MB solutions was analyzed with a pH meter. The batch adsorption experiment was conducted in 300 mL Erlenmeyer flasks with 250 mL solutions of MB added to each flask at an initial concentration range of 25 to 500 mg/L. Samples (0.125 g) with a particle size of <0.25 mm were put into the MB solution (250 mL) and agitated in a shaker (150 rpm) at 25 °C. For the equilibrium studies, the batch was run for 540 min to ensure that equilibrium conditions were achieved.

For the batch adsorption experiment, after certain times (e.g., 15, 30 min, 60 min, . . ., 540 min), 10 mL was taken with a pipette from the supernatant of each flask and centrifuged at 5000 rpm for 6 min in a Hettich EBA 20 centrifuge to separate suspended particles from the prepared material. A small amount (2 mL) of supernatant solution was obtained from each flask, diluted with suitable amounts of deionized water, and then stored in a sealed bottle for analysis using a UV–Vis spectrophotometer (Jasco V-770, Oklahoma City, OK, USA). The remaining 8 mL was returned to the flask to avoid loss of adsorbance. The concentration of MB was established by comparing the adsorbance with a calibration curve. Equation (1) was used to determine the total MB adsorbed, $q_t$ (mg/g), onto the CMs at time $t$:

$$q_t = \frac{(C_o - C_t)\,V}{M} \tag{1}$$

where $q_t$ (mg/g) is the amount of adsorption at time $t$, $C_o$ represents the initial concentration of the solution (mg/L), $C_t$ represents the concentration of the solution at time $t$ (mg/L), $V$ represents the volume of the solution (L), and $M$ represents the mass of the dry adsorbent (g).

The amount of MB adsorbed at equilibrium, $q_e$ (mg/g), was calculated using Equation (2):

$$q_e = \frac{(C_o - C_e)\,V}{M} \tag{2}$$

where $q_e$ (mg/g) represents the amount of adsorption at equilibrium while $C_e$ represents the solution equilibrium concentration (mg/L).

The removal efficiency can be calculated using Equation (3):

$$R(\%) = \frac{(C_o - C_e)}{C_o} \times 100 \tag{3}$$

*2.6. Reusability and Recovery of Adsorbent*

A desorption study, or regeneration study, was conducted to assess the ability of the adsorbate to be removed from the mobile phase. This analysis is important to explain the transfer process of the adsorbate between the aqueous phase and the solid phase. Five cycles of adsorption—desorption were studied in this experimental work. Ethanol was used as a regenerative solution, as MB is soluble in ethanol. The adsorption and desorption process was conducted using a method elucidated by Genli et al. [42] and Al-Awadi et al. [3]. Briefly, three activated samples were separately mixed with 100 mL of ethanol solution in a series of conical flasks, and the mixtures were agitated at 150 rpm for 2 h using a digital orbital shaker (Jeio tech rotary shaker, SI -600, Seoul, Korea). The conical flasks were closed with rubber stoppers to avoid vaporization of the regenerative solution. After the mixing process was completed, the solutions were then separated from the solid particles via centrifuge (Bioevopeak CFG-18.5B, Jinan, China) at 6000 rpm speed for 6 min, followed by the measuring of adsorbance values using a UV–Vis spectrophotometer (UV–Vis, UV5, Mettler Toledo, Greifensee, Switzerland). The adsorbance values (nm) of the filtrates were then recorded. The adsorbent was then dried at 80 °C overnight. The adsorption and desorption cycles of MB on activated materials were carried out by washing with water and ethanol followed by regeneration and then subsequent reuse. The activated samples were regenerated into their original form and reutilized for five adsorption–desorption cycles. The percentage of removal (%) for each cycle in the batch adsorption process was then calculated. The AM sample was tested for the reusability study and multistage adsorption. Experiments were performed with 200 mL of the MB solution and 100 mg of the activated material at room temperature. The operating conditions were 240 min of contact time and an initial concentration of 300 mg/L MB.

**3. Results and Discussions**

*3.1. Formation and Morphologies of Carbon Microspheres*

The yields of the conversions of the three biomass samples—seeds, molasses, and leaflets—were 43%, 32%, and 52%, respectively. The micrographs of CMs for the three samples are presented in Figure 1. Figure 1a shows that CMs were dispersed on the surface of the seed sample at 230 °C 4 h before activation with phosphoric acid. The CMs appear to possess high degrees of sphericity and are well-developed and reasonably homogeneous. As shown in Figure 1a, the largest and smallest CM diameters were 16.6 and 4.7 μm, respectively. The average diameter of the CMs is 8.6 μm. Figure 1a′ shows that the CMs were slightly deformed from their original morphology after the activation process. The largest and smallest CM diameters were 15.7 and 4.4 μm, respectively. However, the average size of the particles remained in the same range at about 8.2 μm. Figure 1b,b′ shows the CMs on the leaflet sample before (NAL) and after activation (AL), respectively. It should be noted that CMs on the leaflets were not very well-developed or homogeneous as compared to those on the seeds and palm date molasses. In addition, the spheres were slightly smaller in size than those of the seeds or palm date molasses. From Figure 1b, it can be seen that the diameter of the smallest CM was 3.1 μm, while the diameter of the largest CM was 11.7 μm. In contrast, for the activated sample (Figure 1b′), the smallest

and largest diameters were 3.1 μm and 9.6 μm, respectively. The average diameter was 5.8 μm. Figure 1c,c' illustrates the CMs developed for the palm date crystallized molasses before (NAM) and after activation (AM), respectively. The CMs appear to be fully formed and very regular. The average sizes of the particles before and after activation were about 7.5 and 7.2 μm respectively. It is important to note that obtaining CMs from cellulosic materials is more complex than obtaining them from saccharides, such as glucose, sucrose, and fructose, due to the compact structure of highly crystallized cellulose. Consequently, CMs formed from leaflet samples required an 8 h residence time, which was longer than the time for CMs obtained from the seed sample and crystallized molasses. Figure 1a',c' shows that the CMs retained their original spherical morphology without any noticeable change in the diameter of the particles. The average sizes of the particles before and after activation were about 8.5 and 8.2 μm, respectively.

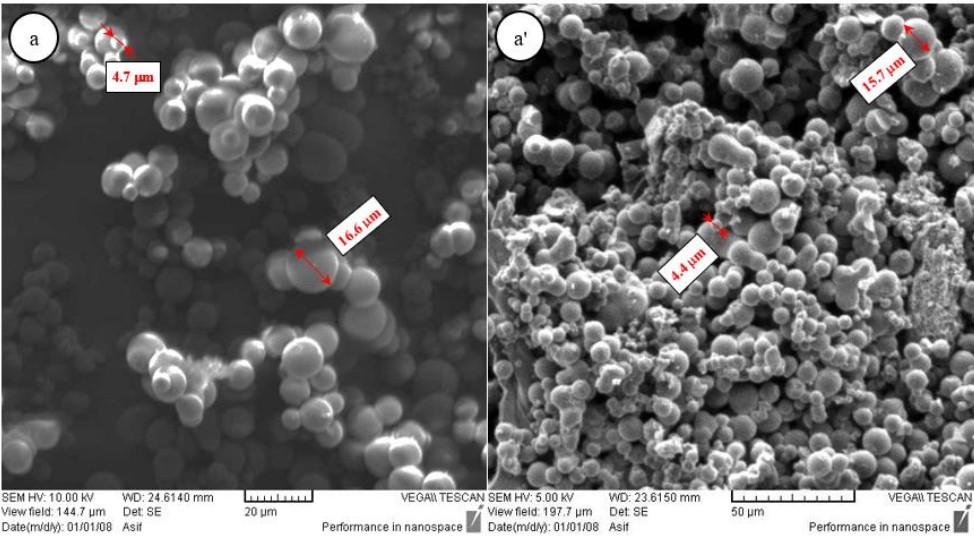

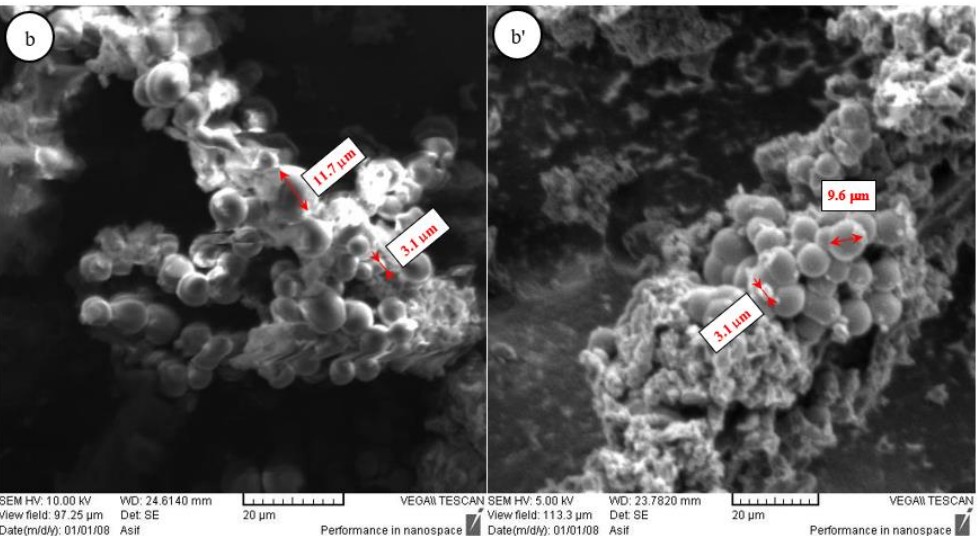

**Figure 1.** *Cont.*

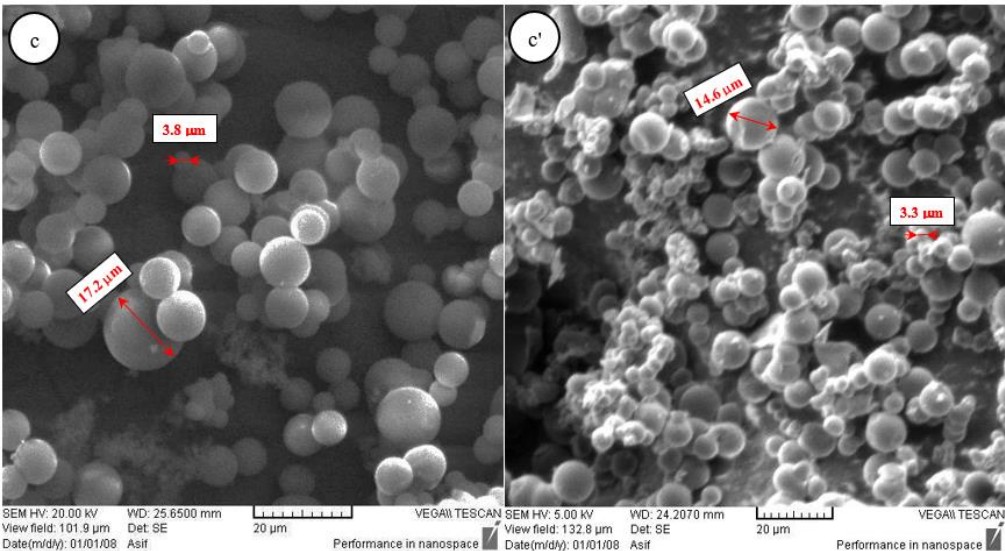

**Figure 1.** Micrographs for (**a**) non-activated seeds (NAS) at 230 °C and 4 h, (**a′**) activated seeds (AS) at 230 °C and 4 h, (**b**) non-activated leaflets (NAL) at 230 °C and 8 h, (**b′**) activated leaflets (AL) at 230 °C and 8 h, (**c**) non-activated molasses (NAM) at 230 °C and 3 h, and (**c′**) activated molasses (AM) at 230 °C and 3 h.

### 3.2. Surface Textural Properties of Carbon Microspheres

Figure 2 shows the N$_2$ adsorption–desorption isotherms of activated and non-activated CM samples. The isotherms for all non-activated samples can be generically classified as type 1 in the IUPAC classification. N$_2$ adsorption on these materials was observed only at a very low relative pressure ($P/P°$), indicative of the predominant microporosity of the samples, even though there was a slight uptake (deviation) at $P/P°$ higher than 0.95. On the other hand, the activated CMs (AS and AL) show a slight presence of mesoporosity in their textures through marginal hysteresis loops—giving them a quasi type IV IUPAC classification. The palpable absence of a plateau at the end of hysteresis implies that it is not solely type IV. The figure on pore size distribution suggests that the increases in pore volume are attributed to the addition of mesoporosity within the 20–50 Angstrom range. The existence of marginal hysteresis loops in the AS and AL samples indicates the existence of peripheral mesopores, which may be promising for adsorption in a solution environment.

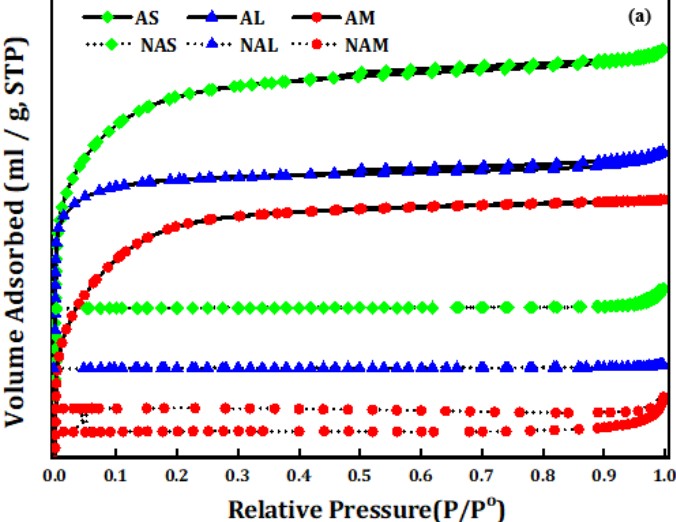

**Figure 2.** *Cont.*

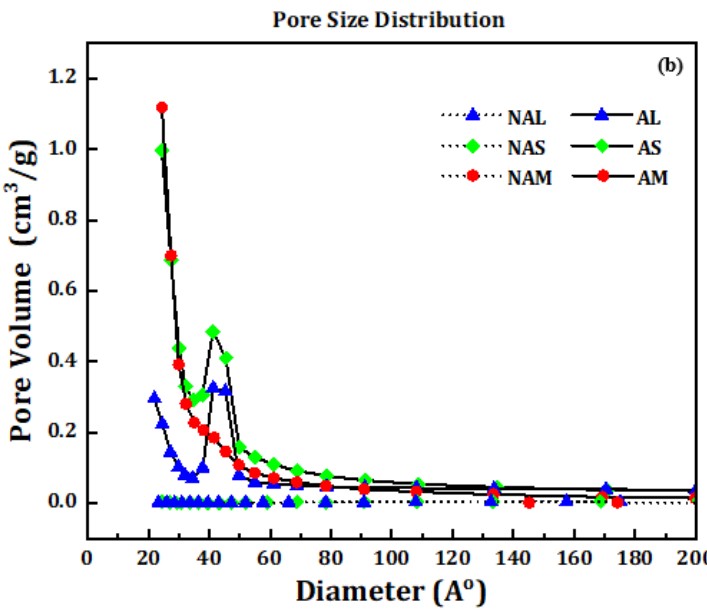

**Figure 2.** (**a**) Nitrogen physisorption isotherms and (**b**) pore size distribution for the inactivated and activated samples.

Table 1 shows the BET surface areas of the three synthesized samples before and after activation. The three samples possessed small surface areas before activation ($<60$ m$^2$/g), indicating that the internal pores on the CM particles were not fully developed. However, upon chemical treatment with phosphoric acid, pores developed on the surface of the microspheres, greatly increasing their surface areas (808–1584 m$^2$/g). Table 1 reveals that the pore sizes of the activated samples increased substantially after activation. In reference to the International Union of Pure and Applied Chemistry (IUPAC) categorization [43], all samples could be classified as mesoporous. The molecular size of MB is 0.95 nm in width and length (1.382 nm–1.447 nm) [44,45]. This indicates that the size of MB molecules is smaller compared to the pore size of all developed adsorbents. Thus, the MB molecules could be easily attached and fitted into the pores of the inactivated and activated materials, which facilitated MB removal from the aqueous solution.

**Table 1.** BET surface area for the inactivated and activated samples.

| CM Samples | Before Activation | | |
|---|---|---|---|
| | BET (m$^2$/g) | Pore Volume (cm$^3$/g) | Pore Size (nm) |
| Non-activated seeds (NAS) | 5.02 | 0.041 | 30.44 |
| Non-activated leaflets (NAL) | 2.21 | 0.0086 | 18.04 |
| Non-activated molasses (NAM) | 0.72 | 0.0033 | 36.66 |
| Activated seeds (AS) | 1584 | 0.47 | 2.52 |
| Activated leaflets (AL) | 808 | 0.156 | 3.33 |
| Activated molasses (AM) | 1543 | 0.48 | 2.22 |

Boehm titration was performed for all prepared samples. The results are shown in Figure 3, indicating that the AM samples contained more phenolic groups that the other samples. Indeed, the phenolic groups dominated the surfaces of both the activated and non-activated samples. The total oxygenated (acidic) functional groups on all samples ranged from 0.1 to 0.4 mmol/g. The elemental analysis results are summarized in Table 2 for all samples. In general, the activated samples possessed higher carbon and lower hydrogen contents than non-activated samples, with the exception of AS. The results also show that AS had a higher carbon content than the other samples. A high oxygen concentration in the AM sample could indicate that many oxygen-containing groups were present in addition to the carboxyl groups [46].

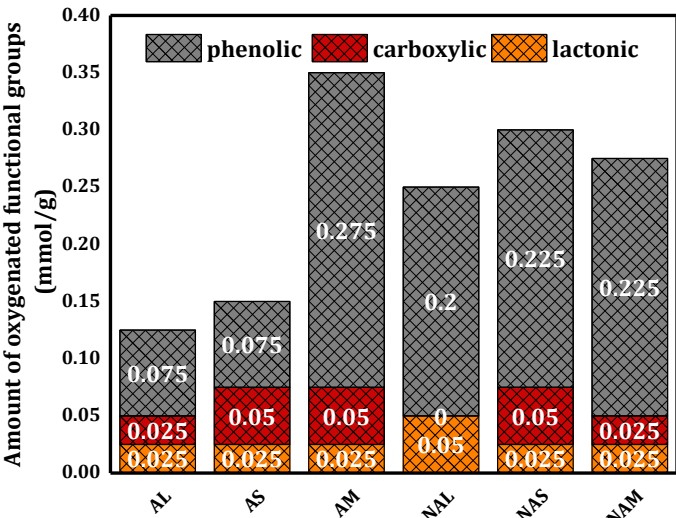

**Figure 3.** The total acidic (oxygenated) functional groups on the surface of the inactivated and activated CM samples.

**Table 2.** Elemental analysis results of the prepared CM materials.

| CM Sample | Chemical Composition | | | | | O/C (Atomic) | H/C (Atomic) |
|---|---|---|---|---|---|---|---|
| | C (wt.%) | H (wt.%) | N (wt.%) | S (wt.%) | O (wt.%) | | |
| NAS | 65.53 ± 0.58 | 4.86 ± 0.09 | 12.65 ± 0.14 | 0.69 ± 0.005 | 16.27 ± 0.16 | 0.19 | 0.89 |
| NAL | 58.99 ± 0.53 | 6.58 ± 0.13 | 21.65 ± 0.24 | 0.92 ± 0.007 | 11.86 ± 0.13 | 0.15 | 1.34 |
| NAM | 67.80 ± 0.60 | 4.81 ± 0.09 | 14.78 ± 0.16 | 0.62 ± 0.005 | 11.99 ± 0.14 | 0.13 | 0.85 |
| AS | 77.48 ± 0.69 | 2.96 ± 0.06 | 11.42 ± 0.12 | 0.36 ± 0.003 | 7.78 ± 0.12 | 0.08 | 0.46 |
| AL | 70.12 ± 0.63 | 2.57 ± 0.05 | 25.35 ± 0.27 | 0.72 ± 0.006 | 1.24 ± 0.01 | 0.01 | 0.44 |
| AM | 49.41 ± 0.44 | 3.57 ± 0.07 | 8.29 ± 0.09 | 0.49 ± 0.004 | 38.24 ± 0.31 | 0.58 | 0.87 |

Variations related to the elemental compositions of the biomass and hydrothermal-based carbon microspheres can be studied using the conventional Van Krevelen diagram (not shown for brevity). From this plot, it appears that the H/C and O/C atomic ratios adhere to the same trend as the dehydration process, with a slight deviation in the direction of the H/C atomic ratio, indicating a low occurrence of the decarboxylation process [47]. The FTIR spectra for the CM samples are shown in Figure 4. Stretching vibrations attributed to aliphatic C–H groups are exhibited at around 2920 cm$^{-1}$ [10]. The bands at approximately 1700 and 1610 cm$^{-1}$ can be assigned to C=O and C=C, respectively, implying the existence of aromatic rings (*ibid*).

### 3.3. Adsorption Studies

#### 3.3.1. Effect of pH

Adsorption efficiency is closely related to pH, since a change in pH can cause the surface charge of an adsorbent to be affected by protonation and deprotonation. The effect of pH on the adsorption process of MB on the three activated samples was examined by varying the pH values from 3 to 8 under the following conditions: an adsorbent dosage of 25 mg, an MB concentration of 300 ppm, a 50 mL MB solution, and a stirring speed of 150 rpm for 2 h. The amount of adsorbed MB, *q*, against pH for the activated samples was plotted, as shown in Figure 5. The uptake of MB enhances with the increase in pH from 3 to 6. The optimum pH of the solution was found to be 6 for all adsorbents, and equilibrium was reached after about 120 min of contact time. The maximum adsorption capacity of MB on the activated CM samples (AS, AL, AM) was 306 ± 8, 223 ± 7, and 393.4 ± 4.7 mg/g, respectively. After a pH of 6, the maximum amounts of MB gradually decrease.

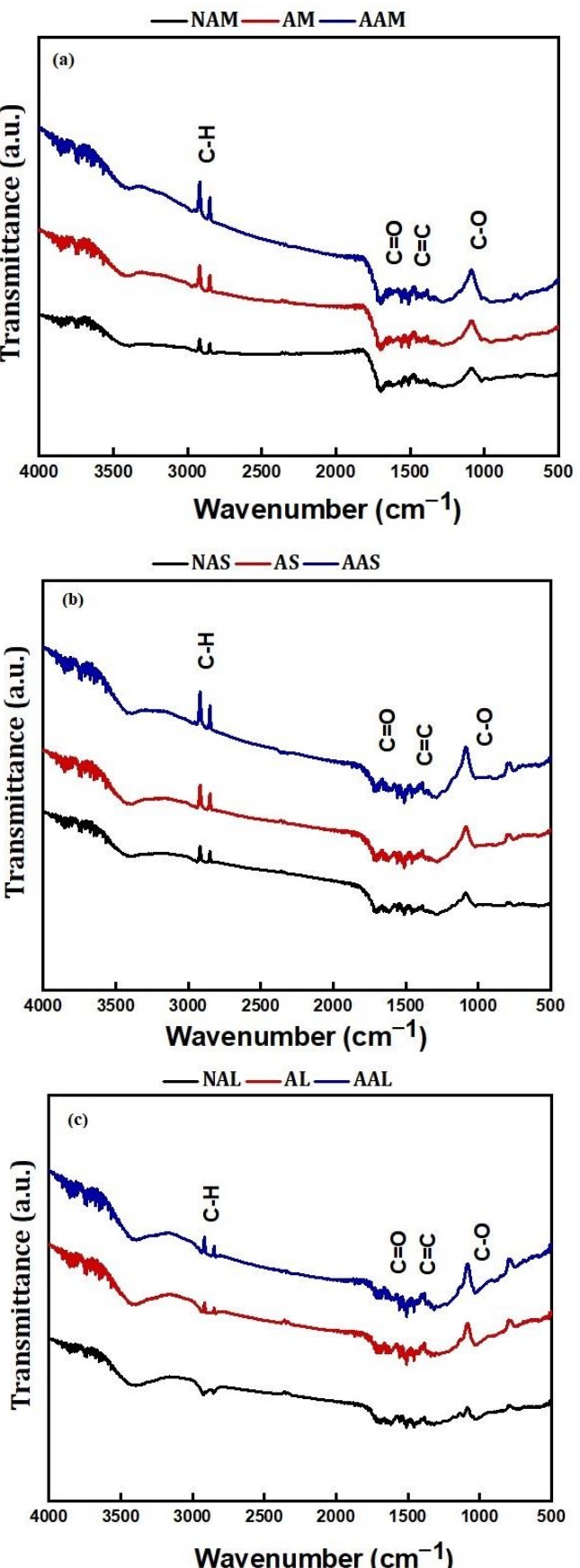

**Figure 4.** FT–IR spectra for (**a**) molasses, (**b**) seeds, and (**c**) leaflets.

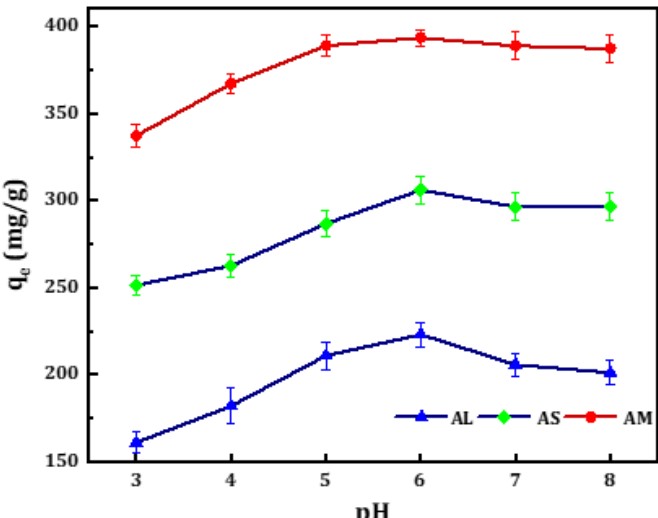

**Figure 5.** The influence of pH on the adsorption of MB on activated samples.

These results are in agreement with the literature, where it has been found that the increase in the adsorption amount of MB with an increase in pH can be explained by the high electrostatic interaction between the negatively charged adsorbent surface and the cationic MB [48–50]. Bharathi et al. [51] found that the removal of cationic dye increases at a high pH due to the high concentration of H+ competing with the cationic groups in MB for adsorption.

### 3.3.2. The Influence of Initial Concentration and Contact Time

The effect of the initial concentration of MB solution on the adsorption capacity of the activated CM samples was investigated at dye concentrations from 25 to 500 mg/L and a fixed contact time of 540 min. The adsorption capacities of the adsorbents were determined. Equilibrium was achieved after 120 min. The removal of MB was faster in the initial stage, then gradually decreased, and became constant after reaching equilibrium. Figure 6 shows the adsorption capacities for MB at 25 °C in activated and activated samples. It should be noted that the inactivated samples (NAS, NAL, and NAM) adsorbed MB only at low concentrations (e.g., 25 and 50 ppm). This can be attributed to differences in the surface area and pore size of the adsorbents, which affect their adsorption capacity. The maximum amounts of MB adsorbed, $q_e$, by NAS, NAL, and NAM at 25 ppm were 2.25, 5.77, and 12.06 mg/g, respectively. Whereas, for the activated samples, the adsorption capacity of the prepared samples increased with an increase in the concentration of MB. The maximum adsorption capacities were observed for AM, 410.4 ± 11.1 mg/g, followed by AS, 327.8 ± 6.7 mg/g, and AL, 249.2 ± 5.4 mg/g. The results showed that the three activated samples' adsorption capacities enhanced with an increase in the initial dye concentration. The error values for the results of inactivated samples are less than 0.5%.

### 3.3.3. Adsorption Isotherm

The adsorption isotherm explains how adsorbent particles interact with adsorbent material, which is vital for optimizing adsorption mechanisms, elucidating surface properties and adsorption capacities, and designing effective adsorption systems. Therefore, various adsorption models were used in their non-linear forms to analyze the obtained data. The program OriginPro—Graphing and Data Analysis was used for non-linear regression and the parameters of the models were determined and compared to establish the adsorption behavior of the studied system.

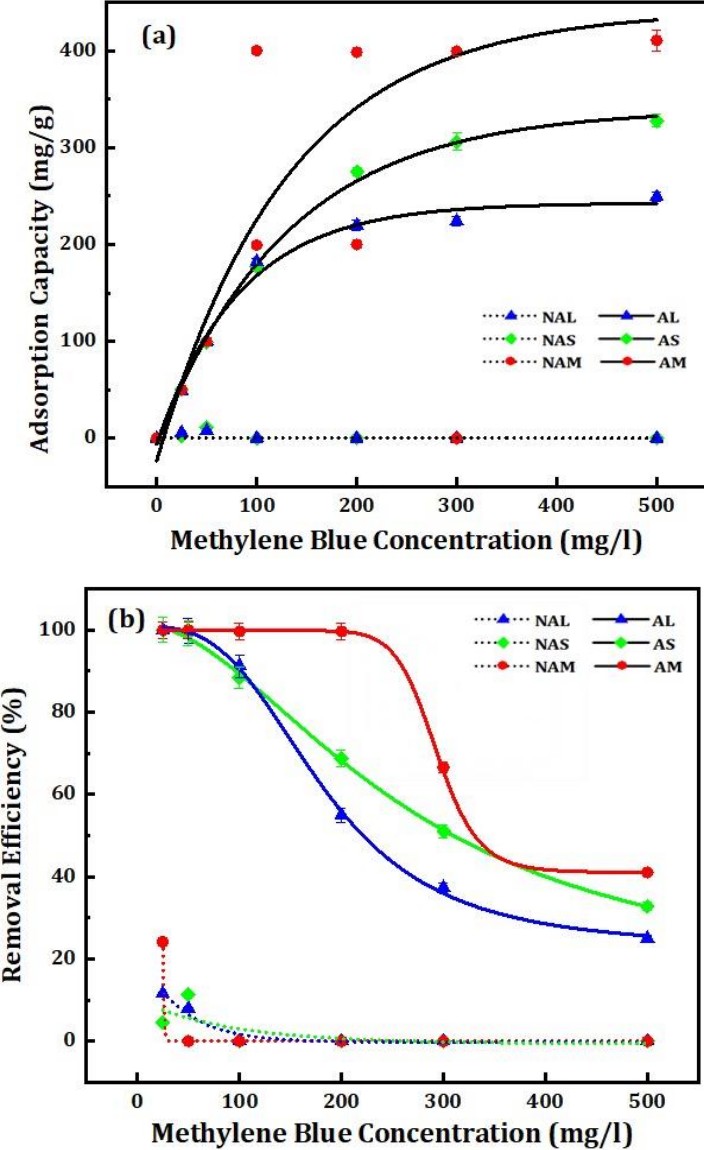

**Figure 6.** (**a**) Adsorption capacities of inactivated and activated samples. (**b**) Removal efficiency of inactivated and activated samples.

Langmuir Isotherm

The Langmuir adsorption model [52] was applied to one-layer adsorption on active locations of the adsorbent (Equation (4)):

$$q_e = \frac{q_m K_L C_e}{1 + K_L C_e} \tag{4}$$

in which $q_m$ (mg/g) represents the amount of adsorbed MB per unit mass of CMs relative to the complete monolayer coverage, and $K$ denotes the Langmuir constant. A non-linear form of this model was conducted and its parameters are listed in Table 3. It can be seen that the experimental data fit the Langmuir isotherm model quite well, with determination coefficients, $R^2$, close to 0.90 for all prepared samples. The maximal monolayer adsorption capacity obtained for the samples was in the order of AM > AS > AL, with AM having the highest $q_m$ (414.6 mg/g).

**Table 3.** Langmuir, Freundlich, and Temkin isotherm constants for the activated samples.

| CM Samples | Langmuir Constants | | | Freundlich Constants | | | Temkin Constants | | |
|---|---|---|---|---|---|---|---|---|---|
| | $Q_{max}$ | $K_L$ | $R^2$ | $1/n$ | $K_F$ | $R^2$ | $B$ | $A$ | $R^2$ |
| AS | 322.54 | 0.124 | 0.903 | 0.178 | 122.10 | 0.967 | 35.76 | 27.81 | 0.964 |
| AL | 219.71 | 11.55 | 0.919 | 0.098 | 138.90 | 0.953 | 16.69 | $57.18 \times 10^2$ | 0.955 |
| AM | 414.63 | 4.54 | 0.880 | 0.054 | 306.20 | 0.842 | 19.59 | $62.16 \times 10^5$ | 0.844 |

In addition, the Langmuir isotherm is linked to a dimensionless constant—the separation factor ($R_L$) (Equation (5)). The values of $R_L$ can be obtained from the results of the experiment, and a plot of the separation factor against the initial concentration ($C_o$) can be generated. The dimensionless constant in the mathematical expression can be written as follows:

$$R_L = \frac{1}{(1 + K_L C_o)} \qquad (5)$$

in which $C_o$ represents the highest initial MB concentration (mg/L). The value of $R_L$ indicates the type of adsorption process: $R_L < 1$, the adsorption is favorable; $R_L = 1$, the adsorption is linear; $R_L > 1$, the adsorption is unfavorable; and $R_L = 0$, the adsorption is irreversible. Values of $R_L$ have been obtained for AS, AL, and AM and are 0.993, 0.980, and 0.0033, respectively, suggesting that adsorption is a favorable process for all ranges of dye concentrations in this experimental work.

Freundlich Isotherm

The Freundlich adsorption model [53] is given by Equation (6):

$$q_e = K_F C_e^{1/n} \qquad (6)$$

where $K_F$ and $n$ are constants, estimated from the slope and intercept of the linear plot of $\log q_e$ versus $\log C_e$ (Equation (6)). The $1/n$ value is observed to be between the range of 0–1, and adsorption becomes more heterogeneous when the value of $1/n$ approaches 0. A steep slope is seen when $1/n$ is closer to 1, and it reflects a higher adsorption capacity at higher equilibrium concentrations that decreases rapidly at a low equilibrium concentration [54]. Fitting the experimental data to a non-linear form of Freundlich adsorption resulted in $R^2$ values of 0.967, 0.953, and 0.842 for AS, AL, and AM, respectively.

Temkin Isotherm

The Temkin isotherm model [55] is expressed by Equation (7):

$$q_e = B \ln(A_T C_e) \qquad (7)$$

where $B = \frac{RT}{b_T}$ $R$ gas constant and $T$ is the study temperature °C, while $b_T$ and $A_T$ are model constants that depend on the system and adsorbent. The Temkin isotherm observes that the heat of adsorption of all molecules in the layer reduces linearly with coverage due to interactions between adsorbate molecules [56]. Unlike the Freundlich isotherm, the Temkin isotherm assumes an unvarying dispersal of binding energies on the surface. Therefore, the Temkin isotherm is often used to model adsorption on homogeneous surfaces, whereas the Freundlich isotherm is more appropriate for heterogeneous surfaces [2]. The constants of the Temkin model were determined and are listed in Table 3. The coefficients of determination were 0.964, 0.955, and 0.844 for AS, AL, and AM, respectively.

In comparison to all $R^2$ values, it was seen that the experimental data fit well with the investigated models for activated seeds and leaflet samples and, to a lesser extent, with the molasses sample. Hence, it can be deduced that monolayer adsorption took place during the batch adsorption process, utilizing the activated samples, as the experimental data fit the Langmuir adsorption isotherm quite well. The data also revealed that dye

molecules in the solution were not being deposited onto the other dye molecules that had been adsorbed onto the adsorptive sites of the activated samples. Table 4 lists a comparison of the maximum adsorption capacities of MB on different adsorbents. It shows that the activated molasses (AM) in this work possessed outsized and superior adsorption capacities as compared with those of other previously reported biomass, which bodes well for its application in dye contaminant removal from wastewater.

**Table 4.** Comparison of the maximum adsorption capacities of MB on different hydrochar-based adsorbents synthesized via the HTC process.

| Adsorbent | $q_{max}$, (mg/g) | Reference |
|---|---|---|
| Coconut shell | 200.01 | [31] |
| Peanut shell | 1368 | [56] |
| Canola stalk | 93.4 | [57] |
| Bamboo | 655.76 | [30] |
| Sewage sludge | 52.56 | [58] |
| Pine wood | 86.7 | [59] |
| Bamboo | 1155.57 | [60] |
| Rice husk | 53.21 | [61] |
| Coffee husk | 418.78 | [32] |
| Filtrasorb 400 | $295 \pm 3$ | [62] |
| Norit | $276 \pm 3$ | [62] |
| Picacarb | $248 \pm 2$ | [62] |
| Palm date molasses-activated hydrochar (AM) | 414.63 | This work |
| Palm date seed-activated hydrochar (AS) | 322.54 | This work |
| Palm date leaflet-activated hydrochar (AL) | 219.71 | This work |

The generally good fit of our adsorption data with all three models, together with the existence of oxygenated functional groups, implies the prominence of adsorption mechanisms pertaining to electrostatic forces and hydrogen bonding. The π-π interactions between the aromatic rings of MB and the surface of the microspheres may also play a significant role [49].

3.3.4. Adsorption Kinetics

An adsorption kinetic analysis was performed to assess the adsorptive capability of the adsorbent as well as to investigate the mechanism of mass transfer and the reaction rate. In addition, this analysis can afford insight into the adsorption pathway and probable mechanisms involved. The pseudo-second-order kinetic model [63] was adopted in this work in order to outline the dynamic of the adsorption process.

The pseudo-second-order model is expressed by Equation (8):

$$q_t = k_2 q_e^2 t / (1 + k_2 q_e t) \tag{8}$$

in which $k_2$ is the rate constant of the second-order reaction (g/mg. min), and $q_e$ and $q_t$ are the amounts of adsorbate (mg/g) adsorbed at equilibrium and time $t$ (min), respectively. The pseudo-second-order model is centered on the notion that the rate-determining step of adsorption is chemisorption relating to valence forces via the sharing or swap of electrons between the adsorbent and adsorbate. Pseudo-second-order constants ($k_2$) were established for the three activated samples and are listed in Table 5. It was observed that the values of the linear regression for AS, AL, and AM were 0.994, 0.985, and 0.9999, respectively. This agrees with Ho and McKay [64], who concluded that, of all adsorption systems that they analyzed, pseudo-second-order (PSO) kinetics provided the best correlation with experimental data. The theoretical, q_theo, and experimental, q_exp, adsorption values were compared and are presented in Table 5.

**Table 5.** Parameters of pseudo-second-order kinetics model established from experimental data for the synthesized CM samples after activation.

| CM Samples | $q_{exp}$ | $q_{theo.}$ | K | $R^2$ |
|---|---|---|---|---|
| AS | $275.1 \pm 4.7$ | $270.78 \pm 2.3$ | 0.0012 | 0.994 |
| AL | $219.6 \pm 5.6$ | $223.77 \pm 3.7$ | 2.5901 | 0.985 |
| AM | $398.5 \pm 0.065$ | $399.62 \pm 0.8$ | 0.0014 | 0.999 |

### 3.4. Multistage Extraction, Reusability, and Recovery Studies

The MB was not completely removed in a single cycle of adsorption. Thus, it is worthwhile to consider a multistage extraction approach. The evolution of the amount of MB along the extraction cycles is depicted in Figure 7A. As expected, after three successive extraction steps, less than 3% of MB the amount was detected. On the other hand, the reusability of AM without any treatment was limited where the surface of sample was saturated by MB, and less than 1% removal was recorded for the second and third time, as shown in Figure 7B. The regenerated adsorbent samples were studied for five adsorption–desorption cycles, as shown in Figure 7C. According to the results, the AM, AS, and AL retained reuse efficiencies of 46.6, 36.2, and 22.01% respectively, from the original concentration in MB removal subsequent to five successive adsorption–desorption cycles.

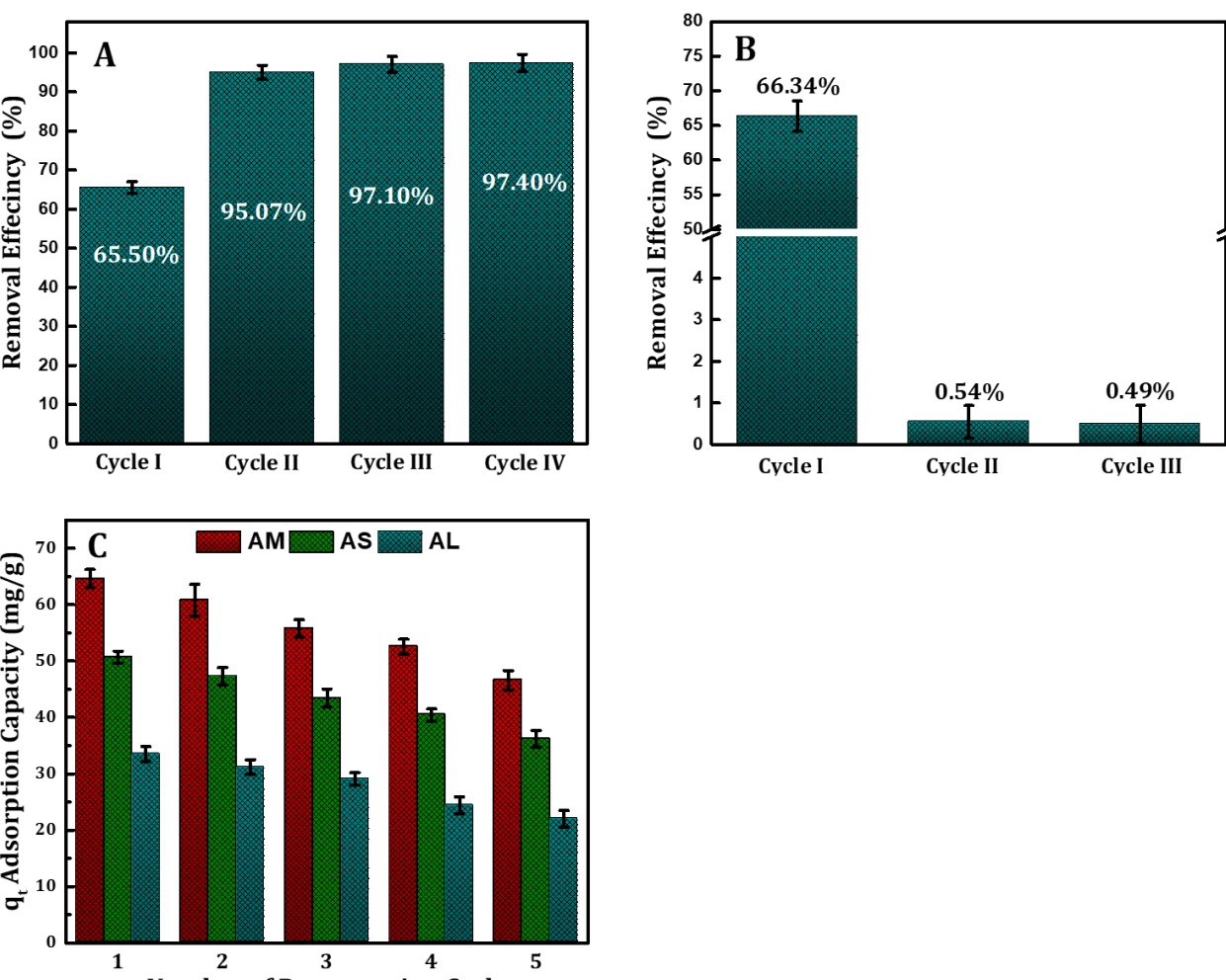

**Figure 7.** Number of cycles and reusability of activated molasses sample (**A**,**B**) and regeneration study of prepared samples (**C**).

## 4. Conclusions

By utilizing the method of hydrothermal carbonization integrated with incipient wetness impregnation, we have synthesized distinctive and comparatively well-defined CMs with outsized BET surface area values from a variety of palm-based biowastes. In terms of surface functionality, phenolic groups dominated the surfaces of both the activated and non-activated samples. Our physisorption analysis suggests that the phosphoric acid-activated process induced higher pore volumes while generating more mesoporosities on the surface of the CMs—such mesoporosities are favorable for liquid-based adsorption processes. The highest MB adsorption capacity, in excess of 400 mg/g and recorded in palm date molasses-activated hydrochar, represents the suitability of our CMs for the adsorption of MB in a fluidic environment. Results from our study are vital because they afford a three-pronged outcome in the form of repurposing and reusing bio-waste resources, the synthesis of valuable microparticles, and applications in dye-based wastewater treatment. The model fitting findings from our work can be employed to scale-up the bench-scale adsorption system for wastewater treatment applications.

**Author Contributions:** Conceptualization, M.E.-H., A.S.A.-A., L.E.B., M.M.A. and C.-Y.Y.; methodology, M.E.-H., A.S.A.-A. and S.A.; validation, L.E.B. and C.-Y.Y.; formal analysis, M.E.-H., A.S.A.-A. and C.-Y.Y.; investigation, L.E.B., A.S.A.-A. and C.-Y.Y.; resources, L.E.B. and S.A.; data curation, C.-Y.Y.; writing—original draft, M.E.-H.; writing—review and editing, C.-Y.Y.; supervision, M.E.-H., L.E.B. and C.-Y.Y.; dunding acquisition, M.E.-H. and M.M.A. All authors have read and agreed to the published version of the manuscript.

**Funding:** This research was funded by the Deputyship for Research and Innovation, "Ministry of Education", in Saudi Arabia for funding this research work through project number IFKSUDR_F158.

**Institutional Review Board Statement:** Not applicable.

**Informed Consent Statement:** Not applicable.

**Data Availability Statement:** Not applicable.

**Acknowledgments:** The authors would like to thank the Deputyship for Research and Innovation, "Ministry of Education", in Saudi Arabia for funding this research work through project number IFKSUDR_F158.

**Conflicts of Interest:** The authors declare no conflict of interest.

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
