# Peer review of "Enhanced Adsorption of Methylene Blue Using Phosphoric Acid-Activated Hydrothermal Carbon Microspheres Synthesized from a Variety of Palm-Based Biowastes"

_coatings, doi:10.3390/coatings13071287_

Round 1

Reviewer 1 Report

The manuscript entitled "Enhanced Adsorption of Methylene Blue using Phosphoric Acid Activated Hydrothermal Carbon Microspheres Synthesized from a Variety of Palm-Based Bio-Wastes" is a good topic and fits the aim and scope of Coatings journal. However, I do not recommend its publication in the present form, unless the authors can revise all the following bullet points.

1.     The numbers on Fig. 1 are not clear.

2.     The authors stated; “novel carbon microspheres (CMs) derived from a date palm (Phoenix dactylifera)”, however, it is a novel at all and there are many papers were synthesized the carbon sphere from date palm!

3.     The unit of the sphere radius in “section 3.1” needs correction.

4.     Adsorption mechanism must be discussed thoroughly. So, I think the following research will be fruitful to the authors; 10.1007/s10934-022-01347-6

5.     What about the selectivity study towards other anionic and cationic dyes.

6.     Fig. 4 does not clarify any FTIR peak.

7.     The adsorbents abbreviations in the comparison study need correction.

8.     The quality of the whole figures needs further improvements.

9.     The explanation of the impact of pH is not accurate, the effect of pH should be clarified based on previous literature and zeta potential. The authors can use the suggested reference in comment 4.

10.  What about the effect of co-existing ions on the adsorption capacity of MB.

English language needs some modifications

Author Response

We wish to thank the reviewers for their insightful comments, which have helped us significantly to improve our manuscript. According to the suggestions, we have thoroughly revised our manuscript and the final version is enclosed. Point-by-point responses to the comments are listed below.

Reviewer #1:

Comments and Suggestions for Authors

The manuscript entitled "Enhanced Adsorption of Methylene Blue using Phosphoric Acid Activated Hydrothermal Carbon Microspheres Synthesized from a Variety of Palm-Based Bio-Wastes" is a good topic and fits the aim and scope of Coatings journal. However, I do not recommend its publication in the present form, unless the authors can revise all the following bullet points.

Comments #1:

The numbers on Fig. 1 are not clear.

Response:

The quality of the numbers in Figure 1 has been improved to make them clearer.

Comments #2:

The authors stated; “novel carbon microspheres (CMs) derived from a date palm (Phoenix dactylifera)”, however, it is not a novel at all and there are many papers were synthesized the carbon sphere from date palm!

Response:

We agree with the reviewer that there are many published papers on the production of hydrochar by the HTC process. However, to the best of our knowledge, there are no previous papers reporting carbon microspheres derived from date palm seeds and leaflets. Only carbon microspheres derived from date palm molasses was reported in our published paper: Materials 2023, 16, 1672. https://doi.org/10.3390/ma16041672. This was cited in the introductory section.

Comments #3:

The unit of the sphere radius in “section 3.1” needs correction.

Response:

We express the size of the microspheres in diameter rather than radius. The unit we used to measure the diameter is mm.

Comments #4:

Adsorption mechanism must be discussed thoroughly. So, I think the following research will be fruitful to the authors; 10.1007/s10934-022-01347-6

Response:

We have added the following discussions at the end of Section 3.3.3:

The generally good fit of our adsorption data to all three models together with the existence of oxygenated functional groups implies the prominence of adsorption mechanisms pertaining to electrostatic forces and hydrogen bonding. The π -π interactions between the aromatic rings of MB and the surface of the microspheres may also play a significant part [49].

Comments #5:

What about the selectivity study towards other anionic and cationic dyes.

Response:

The result of this work refers only to methylene blue. However, the work for other pollutants, which include heavy metals and ionic salts, has not yet been completed. The preliminary results are very promising and will be published as another paper shortly.

Comments #6:

Fig. 4 does not clarify any FTIR peak.

Response:

We have redrawn the FTIR and indicated the most important peaks. Please check Figure 4.

Comments #7:

The adsorbents abbreviations in the comparison study need correction.

Response:

The reviewer is right. We have corrected these abbreviations.

Comments #8:

The quality of the whole figures needs further improvements.

Response:

We have improved the quality of the figures that need improvement.

Comments #9:

The explanation of the impact of pH is not accurate, the effect of pH should be clarified based on previous literature and zeta potential. The authors can use the suggested reference in comment 4.

Response:

The following paragraph is added to section 3.3.1:

“These results are in agreement with the literature where it was found that the increase in the adsorption amount of MB with the increase of pH can be explained by the high electrostatic interaction between the negatively charged adsorbent surface and the cationic MB [48-51]. Bharathi et al. [52] found that the removal of cationic dye increases at high pH due to the high concentration of H+ competing with the cationic groups in MB for adsorption.

Comments #10:

What about the effect of co-existing ions on the adsorption capacity of MB.

Response:

The result of this work refers only to methylene blue. However, the work for other co-existing ions, which include heavy metals and ionic salts, has not yet been completed. The preliminary results are very promising and will be published as another paper shortly.

Comments #11:

Comments on the Quality of English Language

English language needs some modifications

Response:

The manuscript has been thoroughly inspected.

Reviewer 2 Report

The manuscript establishes a protocol for the preparation of new carbon microspheres (CM) using date palm by-products in a hydrothermal carbonization process, activated by phosphoric acid. It characterizes these materials by different techniques and studies the adsorption of methylene blue. The results of the molasses provide by the collecting company and with possibilities of recycling very interesting quantities seems in principle a very innovative and sustainable idea. The kinetic study of the different samples seems to work well, although the equations are over-detailed and are surely within the reach of researchers comparing between CM. However, in the preparation of the materials there are underperformed that are not understood. In principle, the starting product must be washed and cleaned, each sample with different amounts (4g, 2.5g and 4g; seed, leaflets and molasses, respectively), and without knowing what the yield of each preparation is. The result can be very interesting, but if you do not know the cost of each gram of sample, whether seed, leaflets or molasses and its subsequent yield in a previous thermal process (110ºC /12h). Additionally, that it is then activated by addition of phosphoric acid at temperatures above 450ºC, it seems an expensive and inefficient process since the gross yield of this first processing and subsequent activation is unknown.

Regarding the presentation of results, some question:

-In my download the Greek term "mu" does not appear in the text.

-The micrographs together with the figure captions are vague and appear with abbreviations, and the inserts are not visible.

-Table 1 could indicate the samples in name and abbreviations to facilitate the review of the work.

- FT-IR, besides not saying anything significant, nothing is seen, small figure and in this case unimportant. By the way, they do not say how the spectra are made, how many? Is it a pellet, dissolution or ATR?

-Really, I liked the introduction, but the content is not enough to be published in an  journal and should work better on the aspects commented.

Author Response

We wish to thank the reviewers for their insightful comments, which have helped us significantly to improve our manuscript. According to the suggestions, we have thoroughly revised our manuscript and the final version is enclosed. Point-by-point responses to the comments are listed below.

Reviewer #2:

Comments and Suggestions for Authors

The manuscript establishes a protocol for the preparation of new carbon microspheres (CM) using date palm by-products in a hydrothermal carbonization process, activated by phosphoric acid. It characterizes these materials by different techniques and studies the adsorption of methylene blue. The results of the molasses provide by the collecting company and with possibilities of recycling very interesting quantities seems in principle a very innovative and sustainable idea. The kinetic study of the different samples seems to work well, although the equations are over-detailed and are surely within the reach of researchers comparing between CM.

  • However, in the preparation of the materials there are underperformed that are not understood. In principle, the starting product must be washed and cleaned, each sample with different amounts (4g, 2.5g and 4g; seed, leaflets and molasses, respectively).

Response:

We appreciate the reviewer’s favorable comment on our project being a “very innovative and sustainable idea”.

We have added a few lines describing the cleaning and grinding of biomass. Please refer to section 2.1.

  • and without knowing what the yield of each preparation is. The result can be very interesting, but if you do not know the cost of each gram of sample, whether seed, leaflets or molasses and its subsequent yield in a previous thermal process (110ºC /12h).

Response:

We have added a sentence about yield in section 3.1.

  • Additionally, that it is then activated by addition of phosphoric acid at temperatures above 450ºC, it seems an expensive and inefficient process since the gross yield of this first processing and subsequent activation is unknown.

Response:

The activation process with phosphoric acid requires an increase in temperature to this level. This increases the surface area and subsequently increases the adsorption capacity. Without this activation process, the adsorption capacity would be very low.

Regarding the presentation of results, some questions:

Comments #1:

In my download, the Greek term "mu" does not appear in the text.

Response:

This is supposed to be mm and appears fine in the Word version.

Comments #2:

The micrographs together with the figure captions are vague and appear with abbreviations, and the inserts are not visible.

Response:

The abbreviations in Figure 1 have been written in detail and the quality of the numbers in Figure 1 has been improved to make them clearer.

Comments #3:

Table 1 could indicate the samples in name and abbreviations to facilitate the review of the work.

Response:

We have added the full name and abbreviations in Table 1.

Comments #4:

FT-IR, besides not saying anything significant, nothing is seen, small figure and in this case unimportant. By the way, they do not say how the spectra are made, how many? Is it a pellet, dissolution or ATR?

Response:

We have added two lines in section 2.4. We have also redrawn the FTIR figures (please check Figure 4).

Comments #5:

Really, I liked the introduction, but the content is not enough to be published in an journal and should work better on the aspects commented.

Response:

We have improved the introduction and added a paragraph. Please refer to page 2.

Reviewer 3 Report

Coatings 2508257, Review of the manuscript

The manuscript deals with a production of carbon sorbent from Palm-Based Bio-Wastes and shows the possibility to produce efficient sorbents from such wastes.

Table 1. Please, decode abbreviations ones more.

Table 2. What about analysis of CM on phosphorus?

Table 4. Please, insert some data on industrial carbons for the comparison.

Fig. 7c. The ordinate is shown in mg/g, but the comment (line 491) operates with % values.  

Line 132. The following ratios of phosphoric acid/CM sample were investigated: 0.5, 0.75, 1, 1.3, 1.7, 2.3, 3, 4 and 5.

I did not see any dependencies of surface aria or adsorption capacity on the quantity of H3PO4 in the activation process.                                                        

Author Response

We wish to thank the reviewers for their insightful comments, which have helped us significantly to improve our manuscript. According to the suggestions, we have thoroughly revised our manuscript and the final version is enclosed. Point-by-point responses to the comments are listed below.

Comments and Suggestions for Authors

Coatings 2508257, Review of the manuscript

The manuscript deals with the production of carbon sorbent from Palm-Based Bio-Wastes and shows the possibility to produce efficient sorbents from such wastes.

Comments #1:

Table 1. Please, decode abbreviations once more.

Response:

We have added the full name and abbreviations in Table 1.

Comments #2:

Table 2. What about analysis of CM on phosphorus?

Response:

In Table 2, the first three samples (NAS, NAL, and NAM) are inactivated samples. The other three samples (AS, AL, and AM ) are phosphoric acid-activated samples.

Comments #3:

Table 4. Please, insert some data on industrial carbons for comparison.

Response:

A few commercial activated carbon data for the adsorption of MB have been added to Table 4.

Comments #4:

Fig. 7c. The ordinate is shown in mg/g, but the comment (line 491) operates with % values.  

Response:

Figure 7 belongs to Section 3.4 (Multistage Extraction, Reusability, and Recovery Studies), the results of which are usually expressed in %.

Comments #5:

Line 132. The following ratios of phosphoric acid/CM sample were investigated: 0.5, 0.75, 1, 1.3, 1.7, 2.3, 3, 4 and 5. I did not see any dependencies of surface area or adsorption capacity on the quantity of H3PO4 in the activation process.                                                        

Response:

We measured the BET surface area after activation. The ratio of phosphoric acid/CM that gives the highest BET surface area was chosen for the adsorption study. Logically the highest surface area will give the highest adsorption capacity. Therefore, we did not investigate all ratios for the adsorption of MB.

Round 2

Reviewer 1 Report

The manuscript has been carefully revised, so I recommend its publication in the present form.

Reviewer 2 Report

I am satisfied with the corrections incorporated into the manuscript.